# Unsupervised Learning of Object Landmarks through Conditional Image Generation

**Tomas Jakab**[1*]     **Ankush Gupta**[1*]     **Hakan Bilen**[2]     **Andrea Vedaldi**[1]

[1] Visual Geometry Group
University of Oxford
`{tomj,ankush,vedaldi}@robots.ox.ac.uk`

[2] School of Informatics
University of Edinburgh
`hbilen@ed.ac.uk`

## Abstract

We propose a method for learning landmark detectors for visual objects (such as the eyes and the nose in a face) without any manual supervision. We cast this as the problem of generating images that combine the appearance of the object as seen in a first example image with the geometry of the object as seen in a second example image, where the two examples differ by a viewpoint change and/or an object deformation. In order to factorize appearance and geometry, we introduce a tight bottleneck in the geometry-extraction process that selects and distils geometry-related features. Compared to standard image generation problems, which often use generative adversarial networks, our generation task is conditioned on both appearance and geometry and thus is significantly less ambiguous, to the point that adopting a simple perceptual loss formulation is sufficient. We demonstrate that our approach can learn object landmarks from synthetic image deformations or videos, all without manual supervision, while outperforming state-of-the-art unsupervised landmark detectors. We further show that our method is applicable to a large variety of datasets — faces, people, 3D objects, and digits — without any modifications.

## 1  Introduction

There is a growing interest in developing machine learning methods that have little or no dependence on manual supervision. In this paper, we consider in particular the problem of learning, without external annotations, detectors for the landmarks of object categories, such as the nose, the eyes, and the mouth of a face, or the hands, shoulders, and head of a human body.

Our approach learns landmarks by looking at images of deformable objects that differ by acquisition time and/or viewpoint. Such pairs may be extracted from video sequences or can be generated by randomly perturbing still images. Videos have been used before for self-supervision, often in the context of future frame prediction, where the goal is to generate future video frames by observing one or more past frames. A key difficulty in such approaches is the high degree of ambiguity that exists in predicting the motion of objects from past observations. In order to eliminate this ambiguity, we propose instead to condition generation on two images, a source (past) image and a target (future) image. The goal of the learned model is to reproduce the target image, given the source and target images as input. Clearly, without further constraints, this task is trivial. Thus, we pass the target through a tight bottleneck meant to *distil the geometry of the object* (fig. 1). We do so by constraining the resulting representation to encode spatial locations, as may be obtained by an object landmark detector. The source image and the encoded target image are then passed to a generator network which reconstructs the target. Minimising the reconstruction error encourages the model to learn landmark-like representations because landmarks can be used to encode the *geometry* of the object,

---

[*]equal contribution.

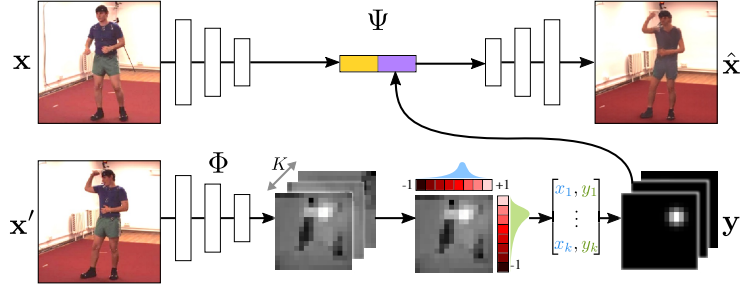

Figure 1: **Model Architecture.** Given a pair of source and target images $(\mathbf{x}, \mathbf{x}')$, the pose-regressor $\Phi$ extracts $K$ heatmaps from $\mathbf{x}'$, which are then marginalized to estimate coordinates of keypoints, to limit the information flow. 2D Gaussians $(\mathbf{y}')$ are rendered from these keypoints and stacked along with the image features extracted from $\mathbf{x}$, to reconstruct the target as $\Psi(\mathbf{x}, \mathbf{y}') = \hat{\mathbf{x}}'$. By restricting the information-flow our model learns semantically meaningful keypoints, without any annotations.

which changes between source and target, while the appearance of the object, which is constant, can be obtained from the source image alone.

The key advantage of our method, compared to other works for unsupervised learning of landmarks, is the simplicity and generality of the formulation, which allows it to work well on data far more complex than previously used in unsupervised learning of object landmarks, *e.g.* landmarks for the highly-articulated human body. In particular, unlike methods such as [45, 44, 55], we show that our method can learn from synthetically-generated image deformations as well as raw videos as it *does not* require access to information about correspondences, optical-flow, or transformation between images.

Furthermore, while image generation has been used extensively in unsupervised learning, especially in the context of (variational) auto-encoders [22] and Generative Adversarial Networks (GANs [13]; see section 2), our approach has a key advantage over such methods. Namely, conditioning on both source and target images simplifies the generation task considerably, making it much easier to learn the generator network [18]. The ensuing simplification means that we can adopt the direct approach of minimizing a perceptual loss as in [10], without resorting to more complex techniques like GANs. Empirically, we show that this still results in excellent image generation results and that, more importantly, semantically consistent landmark detectors are learned without manual supervision (section 4). Project code and details are available at: `http://www.robots.ox.ac.uk/~vgg/research/unsupervised_landmarks/`

## 2   Related work

The recent approaches of [45, 44] learn to extract landmarks based on the principles of equivariance and distinctiveness. In contrast to our work, these methods are not generative. Further, they rely on known correspondences between images obtained either through optical flow or synthetic transformations, and hence, cannot leverage video data directly. Since the principle of equivariance is orthogonal to our approach, it can be incorporated as an additional cue in our method.

Unsupervised learning of representations has traditionally been achieved using auto-encoders and restricted Boltzmann machines [14, 47, 15]. InfoGAN [6] uses GANs to disentangle factors in the data by imposing a certain structure in the latent space. Our approach also works by imposing a latent structure, but using a *conditional*-encoder instead of an auto-encoder.

Learning representations using conditional image generation via a bottleneck was demonstrated by Xue *et al*. [52] in variational auto-encoders, and by Whitney *et al*. [50] using a discrete gating mechanism to combine representations of successive video frames. Denton *et al*. [8] factor the pose and identity in videos through an adversarial loss on the pose embeddings. We instead design our bottleneck to explicitly shape the features to resemble the output of a landmark detector, without any adversarial training. Villegas *et al*. [46] also generate future frames by extracting a representation of appearance and human pose, but, differently from us, require ground-truth pose annotations. Our method essentially *inverts* their analogy network [36] to output landmarks given the source and target image pairs.

Several other generative methods [42, 40, 37, 48, 32] focus on video extrapolation. Srivastava *et al*. [40] employ Long Short Term Memory (LSTM) [16] networks to encode video sequences into fixed-length representation and decode it to reconstruct the input sequence. Vondrick *et al*. [48] propose a GAN for videos, also with a spatio-temporal convolutional architecture that disentangles foreground and background to generate realistic frames. Video Pixel Networks [20] estimate the discrete joint distribution of the pixel values in a video by encoding different modalities such as time, space and colour information. In contrast, we learn a *structured embedding* that explicitly encodes the spatial location of object landmarks.

A series of concurrent works propose similar methods for unsupervised learning of object structure. Shu *et al*. [38] learn to factor a single object-category-specific image into an appearance template in a canonical coordinate system, and a deformation field which warps the template to reconstruct the input, as in an auto-encoder. They encourage this factorisation by controlling the size of the embeddings. Similarly, Wiles *et al*. [51] learn a dense deformation field for faces but obtain the template from a second related image, as in our method. Suwajanakorn *et al*. [43] learn 3D-keypoints for objects from two images which differ by a known 3D transformation, by enforcing equivariance [45]. Finally, the method of Zhang *et al*. [55] shares several similarities with ours, in that they also use image generation with the goal of learning landmarks. However, their method is based on generating a single image from *itself* using landmark-transported features. This, we show is insufficient to learn geometry and requires, as they do, to also incorporate the principle of equivariance [45]. This is a key difference with our method, as ours results in a much simpler system that does *not* require to know the optical-flow/correspondences between images, and can learn from raw videos directly.

## 3    Method

Let $\mathbf{x}, \mathbf{x}' \in \mathcal{X} = \mathbb{R}^{H \times W \times C}$ be two images of an object, for example extracted as frames in a video sequence, or synthetically generated by randomly deforming $\mathbf{x}$ into $\mathbf{x}'$. We call $\mathbf{x}$ the source image and $\mathbf{x}'$ the target image and we use $\Omega$ to denote the image domain, namely the $H \times W$ lattice.

We are interested in learning a function $\Phi(\mathbf{x}) = \mathbf{y} \in \mathcal{Y}$ that captures the "structure" of the object in the image as a set of $K$ object landmarks. As a first approximation, assume that $\mathbf{y} = (u_1, \ldots, u_K) \in \Omega^K = \mathcal{Y}$ are $K$ coordinates $u_k \in \Omega$, one per landmark.

In order to learn the map $\Phi$ in an unsupervised manner, we consider the problem of conditional image generation. Namely, we wish to learn a generator function

$$\Psi : \mathcal{X} \times \mathcal{Y} \to \mathcal{X}, \qquad (\mathbf{x}, \mathbf{y}') \mapsto \mathbf{x}'$$

such that the target image $\mathbf{x}' = \Psi(\mathbf{x}, \Phi(\mathbf{x}'))$ is reconstructed from the *source image* $\mathbf{x}$ and the *representation* $\mathbf{y}' = \Phi(\mathbf{x}')$ of the *target image*. In practice, we learn both functions $\Phi$ and $\Psi$ jointly to minimise the expected reconstruction loss $\min_{\Psi, \Phi} E_{\mathbf{x}, \mathbf{x}'} \left[ \mathcal{L}(\mathbf{x}', \Psi(\mathbf{x}, \Phi(\mathbf{x}'))) \right]$. Note that, if we do not restrict the form of $\mathcal{Y}$, then a trivial solution to this problem is to learn identity mappings by setting $\mathbf{y}' = \Phi(\mathbf{x}') = \mathbf{x}'$ and $\Psi(\mathbf{x}, \mathbf{y}') = \mathbf{y}'$. However, given that $\mathbf{y}'$ has the "form" of a set of landmark detections, the model is strongly encouraged to learn those. This is explained next.

### 3.1    Heatmaps bottleneck

In order for the model $\Phi(\mathbf{x})$ to learn to extract keypoint-like structures from the image, we terminate the network $\Phi$ with a layer that forces the output to be akin to a set of $K$ keypoint detections. This is done in three steps. First, $K$ heatmaps $S_u(\mathbf{x}; k), u \in \Omega$ are generated, one for each keypoint $k = 1, \ldots, K$. These heatmaps are obtained in parallel as the channels of a $\mathbb{R}^{H \times W \times K}$ tensor using a standard convolutional neural network architecture. Second, each heatmap is renormalised to a probability distribution via (spatial) $\mathrm{Softmax}$ and condensed to a point by computing the (spatial) expected value of the latter:

$$u_k^*(\mathbf{x}) = \frac{\sum_{u \in \Omega} u e^{S_u(\mathbf{x}; k)}}{\sum_{u \in \Omega} e^{S_u(\mathbf{x}; k)}} \tag{1}$$

Third, each heatmap is replaced with a Gaussian-like function centred at $u_k^*$ with a small fixed standard deviation $\sigma$:

$$\Phi_u(\mathbf{x}; k) = \exp\left(-\frac{1}{2\sigma^2}\|u - u_k^*(\mathbf{x})\|^2\right) \tag{2}$$

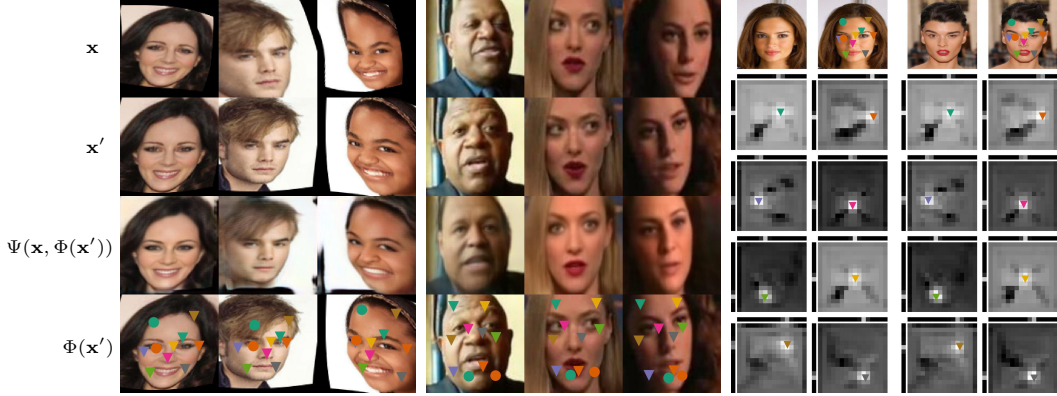

Figure 2: **Unsupervised Landmarks. [left]:** CelebA images showing the synthetically transformed source $\mathbf{x}$ and target $\mathbf{x}'$ images, the reconstructed target $\Psi(\mathbf{x}, \Phi(\mathbf{x}'))$, and the unsupervised landmarks $\Phi(\mathbf{x}')$. **[middle]:** The same for video frames from VoxCeleb. **[right]:** Two example images with selected (8 out of 10) landmarks $u_k$ overlaid and their corresponding 2D score maps $S_u(\mathbf{x}; k)$ (see section 3.1; brighter pixels indicate higher confidence).

The end result is a new tensor $\mathbf{y} = \Phi(\mathbf{x}) \in \mathbb{R}^{H \times W \times K}$ that encodes as Gaussian heatmaps the location of $K$ maxima. Since it is possible to recover the landmark locations exactly from these heatmaps, this representation is equivalent to the one considered above (2D coordinates); however, it is more useful as an input to a generator network, as discussed later.

One may wonder whether this construction can be simplified by removing steps two and three and simply consider $S(\mathbf{x})$ (possibly after re-normalisation) as the output of the encoder $\Phi(\mathbf{x})$. The answer is that these steps, and especially eq. (1), ensure that very little information from $\mathbf{x}$ is retained, which, as suggested above, is key to avoid degenerate solutions. Converting back to Gaussian landmarks in eq. (2), instead of just retaining 2D coordinates, ensures that the representation is still utilisable by the generator network.

**Separable implementation.** In practice, we consider a separable variant of eq. (1) for computational efficiency. Namely, let $u = (u_1, u_2)$ be the two components of each pixel coordinate and write $\Omega = \Omega_1 \times \Omega_2$. Then we set

$$u_{ik}^*(\mathbf{x}) = \frac{\sum_{u_i \in \Omega_i} u_i e^{S_{u_i}(\mathbf{x}; k)}}{\sum_{u_i \in \Omega_i} e^{S_{u_i}(\mathbf{x}; k)}}, \qquad S_{u_i}(\mathbf{x}; k) = \sum_{u_j \in \Omega_j} S_{(u_1, u_2)}(\mathbf{x}; k),$$

where $i = 1, 2$ and $j = 2, 1$ respectively. Figure 2 visualizes the source $\mathbf{x}$, target $\mathbf{x}'$ and generated $\Psi(\mathbf{x}, \Phi(\mathbf{x}'))$ images, as well as $\mathbf{x}'$ overlaid with the locations of the unsupervised landmarks $\Phi(\mathbf{x}')$. It also shows the heatmaps $S_u(\mathbf{x}; k)$ and marginalized separable softmax distributions on the top and left of each heatmap for $K = 10$ keypoints.

## 3.2 Generator network using a perceptual loss

The goal of the generator network $\hat{\mathbf{x}}' = \Psi(\mathbf{x}, \mathbf{y}')$ is to map the source image $\mathbf{x}$ and the distilled version $\mathbf{y}'$ of the target image $\mathbf{x}'$ to a reconstruction of the latter. Thus the generator network is optimised to minimise a reconstruction error $\mathcal{L}(\mathbf{x}', \hat{\mathbf{x}}')$. The design of the reconstruction error is important for good performance. Nowadays the standard practice is to learn such a loss function using adversarial techniques, as exemplified in numerous variants of GANs. However, since the goal here is not generative modelling, but rather to induce a representation $\mathbf{y}'$ of the object geometry for reconstructing a *specific* target image (as in an auto-encoder), a simpler method may suffice.

Inspired by the excellent results for photo-realistic image synthesis of [4], we resort here to use the "content representation" or "perceptual" loss used successfully for various generative networks [12, 1, 9, 19, 27, 30, 31]. The perceptual loss compares a set of the activations extracted from multiple layers of a deep network for both the reference and the generated images, instead of the only raw pixel values. We define the loss as $\mathcal{L}(\mathbf{x}', \hat{\mathbf{x}}') = \sum_l \alpha_l \|\Gamma_l(\mathbf{x}') - \Gamma_l(\hat{\mathbf{x}}')\|_2^2$, where $\Gamma(\mathbf{x})$ is an off-the-shelf pre-trained neural network, for example VGG-19 [39], $\Gamma_l$ denotes the output of the $l$-th sub-network (obtained by chopping $\Gamma$ at layer $l$). As our goal is to have a purely-unsupervised learning, we pre-train the network by using a self-supervised approach, namely colorising grayscale images [25].

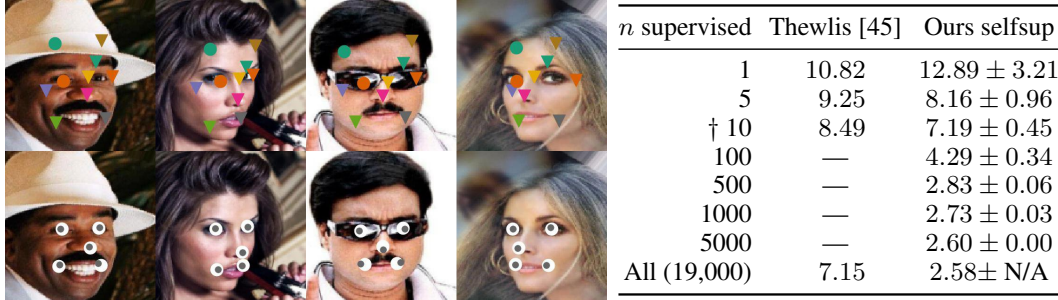

| $n$ supervised | Thewlis [45] | Ours selfsup |
|---|---|---|
| 1 | 10.82 | $12.89 \pm 3.21$ |
| 5 | 9.25 | $8.16 \pm 0.96$ |
| † 10 | 8.49 | $7.19 \pm 0.45$ |
| 100 | — | $4.29 \pm 0.34$ |
| 500 | — | $2.83 \pm 0.06$ |
| 1000 | — | $2.73 \pm 0.03$ |
| 5000 | — | $2.60 \pm 0.00$ |
| All (19,000) | 7.15 | $2.58\pm$ N/A |

Figure 3: **Sample Efficiency for Supervised Regression on MAFL. [left]:** Supervised linear regression of 5 keypoints (bottom-row) from 10 unsupervised (top-row) on MAFL test set. Centre of the white-dots correspond to the ground-truth location, while the dark ones are the predictions. Both unsupervised and supervised landmarks show a good degree of equivariance with respect to head rotation (columns 2, 4) and invariance to headwear or eyewear (columns 1, 3). **[right]:** MSE ($\pm\sigma$) (normalised by inter-ocular distance (in %)) on the MAFL test-set for varying number ($n$) of supervised samples from MAFL training set used for learning the regressor from 30 unsupervised landmarks. †: we outperform the previous state-of-the-art [45] with only 10 labelled examples.

We also test using a VGG-19 model pre-trained for image classification in ImageNet. All other networks are trained from scratch. The parameters $\alpha_l > 0$, $l = 1, \ldots, n$ are scalars that balance the terms. We use a linear combination of the reconstruction error for 'input', 'conv1_2', 'conv2_2', 'conv3_2', 'conv4_2' and 'conv5_2' layers of VGG-19; $\{\alpha_l\}$ are updated online during training to normalise the expected contribution from each layer as in [4]. However, we use the $\ell_2$ norm instead of their $\ell_1$, as it worked better for us.

## 4   Experiments

In section 4.1 we provide the details of the landmark detection and generator networks; a common architecture is used across all datasets. Next, we evaluate landmark detection accuracy on faces (section 4.2) and human-body (section 4.3). In section 4.4 we analyse the invariance of the learned landmarks to various nuisance factors, and finally in section 4.5 study the factorised representation of object style and geometry in the generator.

### 4.1   Model details

**Landmark detection network.**   The landmark detector ingests the image $\mathbf{x}'$ to produce $K$ landmark heatmaps $\mathbf{y}'$. It is composed of sequential blocks consisting of two convolutional layers each. All the layers use $3\times3$ filters, except the first one which uses $7\times7$. Each block doubles the number of feature channels in the previous block, with 32 channels in the first one. The first layer in each block, except the first block, downsamples the input tensor using stride 2 convolution. The spatial size of the final output, outputting the heatmaps, is set to $16\times16$. Thus, due to downsampling, for a network with $n - 3$, $n \geq 4$ blocks, the resolution of the input image is $H \times W = 2^n \times 2^n$, resulting in $16\times16\times(32 \cdot 2^{n-3})$ tensor. A final $1\times1$ convolutional layer maps this tensor to a $16\times16\times K$ tensor, with one layer per landmark. As described in section 3.1, these $K$ feature channels are then used to render $16\times16\times K$ 2D-Gaussian maps $\mathbf{y}'$ (with $\sigma = 0.1$).

**Image generation network.**   The image generator takes as input the image $\mathbf{x}$ and the landmarks $\mathbf{y}' = \Phi(\mathbf{x}')$ extracted from the second image in order to reconstruct the latter. This is achieved in two steps: first, the image $\mathbf{x}$ is encoded as a feature tensor $\mathbf{z} \in \mathbb{R}^{16\times16\times C}$ using a convolutional network with exactly the same architecture as the landmark detection network except for the final $1\times1$ convolutional layer, which is omitted; next, the features $\mathbf{z}$ and the landmarks $\mathbf{y}'$ are stacked together (along the channel dimension) and fed to a regressor that reconstructs the target frame $\mathbf{x}'$.

The regressor also comprises of sequential blocks with two convolutional layers each. The input to each successive block, except the first one, is upsampled two times through bilinear interpolation, while the number of feature channels is halved; the first block starts with 256 channels, and a minimum of 32 channels are maintained till a tensor with the same spatial dimensions as $\mathbf{x}'$ is obtained. A final convolutional layer regresses the three RGB channels with no non-linearity. All

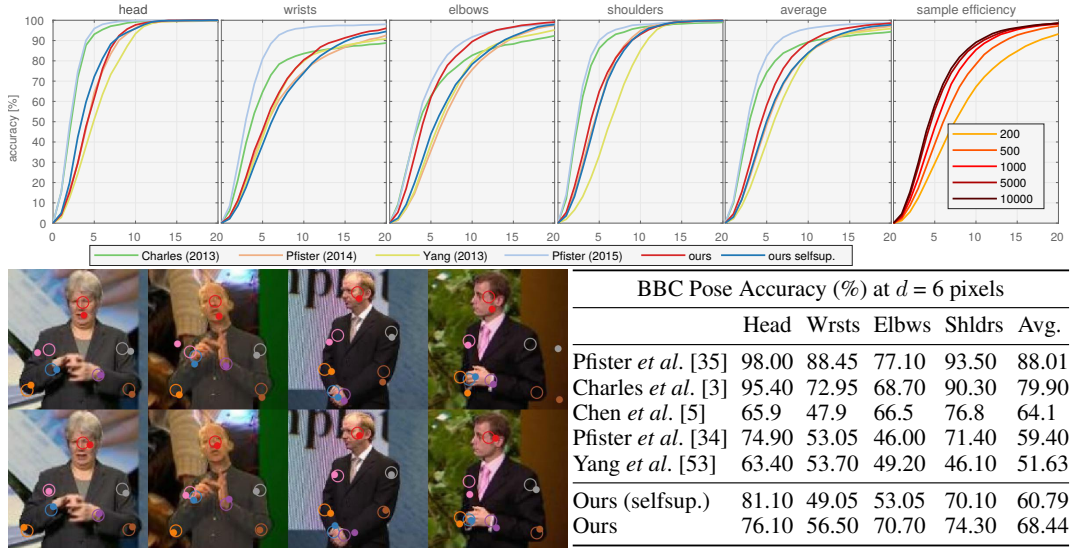

| | BBC Pose Accuracy (%) at $d = 6$ pixels | | | | |
|---|---|---|---|---|---|
| | Head | Wrsts | Elbws | Shldrs | Avg. |
| Pfister *et al.* [35] | 98.00 | 88.45 | 77.10 | 93.50 | 88.01 |
| Charles *et al.* [3] | 95.40 | 72.95 | 68.70 | 90.30 | 79.90 |
| Chen *et al.* [5] | 65.9 | 47.9 | 66.5 | 76.8 | 64.1 |
| Pfister *et al.* [34] | 74.90 | 53.05 | 46.00 | 71.40 | 59.40 |
| Yang *et al.* [53] | 63.40 | 53.70 | 49.20 | 46.10 | 51.63 |
| Ours (selfsup.) | 81.10 | 49.05 | 53.05 | 70.10 | 60.79 |
| Ours | 76.10 | 56.50 | 70.70 | 74.30 | 68.44 |

Figure 4: **Learning Human Pose.** 50 unsupervised keypoints are learnt on the BBC Pose dataset. Annotations (empty circles in the images) for 7 keypoints are provided, corresponding to — head, wrists, elbows and shoulders. Solid circles represent the predicted positions; in **[fig-top]** these are raw discovered keypoints which correspond maximally to each annotation; in **[fig-bottom]** these are regressed (linearly) from the discovered keypoints. **[table]:** Comparison against *supervised* methods; %-age of points within $d = 6$-pixels of ground-truth is reported. **[top-row]:** accuracy-vs-distance $d$, for each body-part; **[top-row-rightmost]:** average accuracy for varying number of supervised samples used for regression.

layers use $3 \times 3$ filters and each block has two layers similarly to the landmark network. All the weights are initialised with random Gaussian noise ($\sigma = 0.01$), and optimised using Adam [21] with a weight decay of $5 \cdot 10^{-4}$. The learning rate is set to $10^{-2}$, and lowered by a factor of 10 once the training error stops decreasing; the $\ell_2$-norm of the gradients is bounded to 1.0.

## 4.2    Learning facial landmarks

**Setup.**    We explore extracting source-target image pairs $(\mathbf{x}, \mathbf{x}')$ using either (1) synthetic transformations, or (2) videos. In the first case, the pairs are obtained as $(\mathbf{x}, \mathbf{x}') = (g_1\mathbf{x}_0, g_2\mathbf{x}_0)$ by applying two random thin-plate-spline (TPS) [11, 49] warps $g_1, g_2$ to a given sample image $\mathbf{x}_0$. We use the 200k CelebA [24] images after resizing them to $128 \times 128$ resolution. The dataset provides annotations for 5 facial landmarks — eyes, nose and mouth corners, which we *do not* use for training. Following [45] we exclude the images in MAFL [57] test-set from the training split and generate synthetically-deformed pairs as in [45, 55], but the transformations themselves are not required for training. We discount the reconstruction loss in the regions of the warped image which lie outside the original image to avoid modelling irrelevant boundary artefacts.

In the second case, $(\mathbf{x}, \mathbf{x}')$ are two frames sampled from a video. We consider VoxCeleb [28], a large dataset of face tracks, consisting of 1251 celebrities speaking over 100k English language utterances. We use the standard training split and remove any overlapping identities which appear in the test sets of MAFL and AFLW. Pairs of frames from the same video, but possibly belonging to different utterances are randomly sampled for training. By using video data for training our models we eliminate the need for engineering synthetic data.

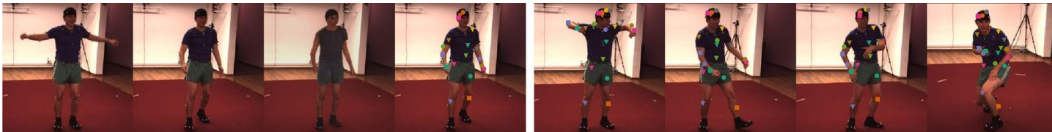

Figure 5: **Unsupervised Landmarks on Human3.6M. [left]:** an example quadruplet source-target-reconstruction-keypoint (left to right) from Human3.6M. **[right]:** learned keypoints on a test video sequence. The landmarks consistently track the legs, arms, torso and head across frames.

**Qualitative results.** Figure 2 shows the learned heatmaps and source-target-reconstruction-keypoints quadruplets $\langle \mathbf{x}, \mathbf{x}', \Psi(\mathbf{x}, \Phi(\mathbf{x}')), \Phi(\mathbf{x}') \rangle$ for synthetic transformations and videos. We note that the method extracts keypoints which consistently track facial features across deformation and identity changes (*e.g.*, the green circle tracks the lower chin, and the light blue square lies between the eyes). The regressed semantic keypoints on the MAFL test set are visualised in fig. 3, where they are localised with high accuracy. Further, the target image $\mathbf{x}'$ is also reconstructed accurately.

**Quantitative results.** We follow [45, 44] and use unsupervised keypoints learnt on CelebA and VoxCeleb to regress manually-annotated keypoints in the MAFL and AFLW [23] test sets. We freeze the parameters of the unsupervised detector network ($\Phi$) and learn a *linear* regressor (without bias) from our unsupervised keypoints to 5 manually-labelled ones from the respective training sets. Model selection is done using $10\%$ validation split of the training data.

We report results in terms of standard MSE normalised by the inter-ocular distance expressed as a percentage [57], and show a few regressed keypoints in fig. 3. Before evaluating on AFLW, we finetune our networks pre-trained on CelebA or VoxCeleb on the AFLW training set. We do not use any labels during finetuning.

*Sample efficiency.* Figure 3 reports the performance of detectors trained on CelebA as a function of the number $n$ of supervised examples used to translate from unsupervised to supervised keypoints. We note that $n = 10$ is already sufficient for results comparable to the previous state-of-the-art (SoA) method of Thewlis *et al.* [45], and that performance almost saturates at $n = 500$ (vs. 19,000 available training samples).

*Vs. SoA.* Table 1 compares our regression results to the SoA. We experiment regressing from $K = \{10, 30, 50\}$ unsupervised landmarks, using the self-supervised and the supervised perceptual loss networks; the number of samples $n$ used for regression is maxed out ($= 19000$) to be consistent with previous works. On both MAFL and AFLW datasets, at $2.58\%$ and $6.31\%$ error respectively (for $K = 30$), we significantly outperform all the supervised and unsupervised methods. Notably, we perform better than the concurrent work of Zhang *et al.* [55] (MAFL: $3.16\%$; AFLW: $6.58\%$), while using a simpler method. When synthetic warps are removed from [55], so that the *equivariance constraint cannot be employed*, our method is significantly better ($2.58\%$ vs

| Method | $K$ | MAFL | AFLW |
|---|---|---|---|
| *Supervised* | | | |
| RCPR [2] | | – | 11.60 |
| CFAN [54] | | 15.84 | 10.94 |
| Cascaded CNN [41] | | 9.73 | 8.97 |
| TCDCN [57] | | 7.95 | 7.65 |
| RAR [41] | | – | 7.23 |
| MTCNN [56] | | 5.39 | 6.90 |
| *Unsupervised / self-supervised* | | | |
| Thewlis [45] | 30 | 7.15 | – |
| | 50 | 6.67 | 10.53 |
| Thewlis [44](frames) | – | 5.83 | 8.80 |
| Shu † [38] | – | 5.45 | – |
| Zhang [55] | 10 | 3.46 | 7.01 |
| w/ equiv. | 30 | 3.16 | 6.58 |
| w/o equiv. | 30 | 8.42 | – |
| Wiles ‡ [51] | – | 3.44 | – |
| *Ours, training set: CelebA* | | | |
| loss-net: selfsup. | 10 | 3.19 | 6.86 |
| | 30 | 2.58 | **6.31** |
| | 50 | **2.54** | 6.33 |
| loss-net: sup. | 10 | 3.32 | 6.99 |
| | 30 | 2.63 | 6.39 |
| | 50 | 2.59 | 6.35 |
| *Ours, training set: VoxCeleb* | | | |
| loss-net: selfsup. | 30 | 3.94 | 6.75 |
| w/ bias | 30 | 3.63 | – |
| loss-net: sup. | 30 | 4.01 | 7.10 |

Table 1: **Comparison with state-of-the-art on MAFL and AFLW.** $K$ is the number of unsupervised landmarks. †: train a 2-layer MLP instead of a *linear* regressor. ‡: use the larger VoxCeleb2 [7] dataset for unsupervised training, and include a bias term in their regressor (through batch-normalization). Normalised %-MSE is reported (see fig. 3).

$8.42\%$ on MAFL). We are also significantly better than many SoA *supervised* detectors [54, 41, 57] using only $n = 100$ supervised training examples, which shows that the approach is very effective at exploiting the unlabelled data. Finally, training with VoxCeleb video frames degrades the performance due to domain gap; including a bias in the linear regressor improves the performance.

| fc-layer ($d$) → | 10 | 20 | 60 | ours $K=30$ | loss → | $\ell_1$ | adv.+ $\ell_1$ | $\ell_2$ | adv.+ $\ell_2$ | content (ours) |
|---|---|---|---|---|---|---|---|---|---|---|
| MAFL | 20.60 | 21.94 | 28.96 | **2.58** | MAFL ($K=30$) | 3.64 | 3.62 | 2.84 | 2.80 | **2.58** |

Table 2: **Abalation Study. [left]:** The keypoint bottleneck when replaced with a low $d$-dimensional, $d = \{10, 20, 60\}$, *fully-connected* (fc) layer leads to significantly worse landmark detection performance (%-MSE) on the MAFL dataset. **[right]:** Replacing the *content* loss with $\ell_1, \ell_2$ losses on the images, optionally paired with an *adversarial* loss (*adv.*) also degrades the performance.

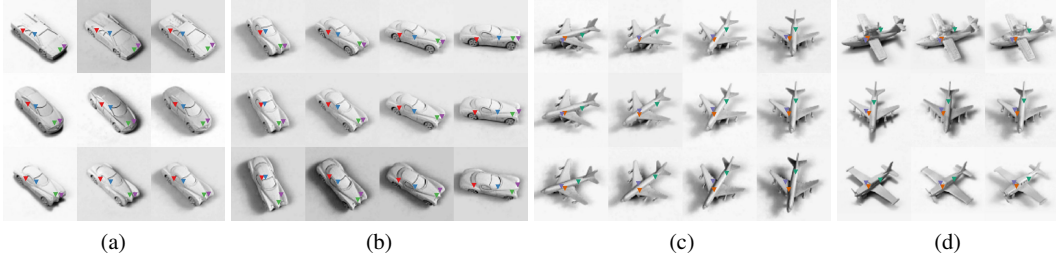

| (a) | (b) | (c) | (d) |

Figure 6: **Invariant Localisation.** Unsupervised keypoints discovered on smallNORB test set for the *car* and *airplane* categories. Out of 20 learned keypoints, we show the most geometrically stable ones: they are invariant to pose, shape, and illumination. **[b–c]:** elevation-vs-azimuth; **[a, d]:** shape-vs-illumination ($y$-axis-vs-$x$-axis).

**Ablation study.** In table 2 we present two ablation studies, first on the keypoint bottleneck, and second where we compare against adversarial and other image-reconstruction losses. For both the settings, we take the best performing model configuration for facial landmark detection on the MAFL dataset.

*Keypoint bottleneck.* The keypoint bottleneck has two functions: (1) it provides a differentiable and distributed representation of the location of landmarks, and (2) it restricts the information from the target image to spatial locations only. When the bottleneck is replaced with a generic low dimensional fully-connected layer (as in a conventional auto-encoder) the performance degrades significantly This is because the continuous vector embedding is not encouraged to encode geometry explicitly.

*Reconstruction loss.* We replace our content/perceptual loss with $\ell_1$ and $\ell_2$ losses on generated pixels; the losses are also optionally paired with an *adversarial* term [13] to encourage verisimilitude as in [18]. All of these alternatives lead to worse landmark detection performance (table 2). While GANs are useful for aligning image distributions, in our setting we reconstruct a *specific* target image (similar to an auto-encoder). For this task, it is enough to use a simple content/perceptual loss.

### 4.3 Learning human body landmarks

**Setup.** Articulated limbs make landmark localisation on human body significantly more challenging than faces. We consider two *video* datasets, BBC-Pose [3], and Human3.6M [17]. BBC-Pose comprises of 20 one-hour long videos of sign-language signers with varied appearance, and dynamic background; the test set includes 1000 frames. The frames are annotated with 7 keypoints corresponding to head, wrists, elbows, and shoulders which, as for faces, we use only for quantitative evaluation, not for training. Human3.6M dataset contains videos of 11 actors in various poses, shot from multiple viewpoints. Image pairs are extracted by randomly sampling frames from the same video sequence, with the additional constraint of maintaining the time difference within the range 3-30 frames for Human3.6M. Loose crops around the subjects are extracted using the provided annotations and resized to $128{\times}128$ pixels. Detectors for $K = 20$ and $K = 50$ keypoints are trained on Human3.6M and BBC-Pose respectively.

**Qualitative results.** Figure 4 shows raw unsupervised keypoints and the regressed semantic ones on the BBC-Pose dataset. For each annotated keypoint, a maximally matching unsupervised keypoint is identified by solving bipartite linear assignment using mean distance as the cost. Regressed keypoints consistently track the annotated points. Figure 5 shows $\langle \mathbf{x}, \mathbf{x}', \Psi\left(\mathbf{x}, \Phi(\mathbf{x}')\right), \Phi(\mathbf{x}')\rangle$ quadruplets, as for faces, as well as the discovered keypoints. All the keypoints lie on top of the human actors, and consistently track the body across identities and poses. However, the model cannot discern frontal and dorsal sides of the human body apart, possibly due to weak cues in the images, and no explicit constraints enforcing such consistency.

**Quantitative results.** Figure 4 compares the accuracy of localising the 7 keypoints on BBC-Pose against *supervised* methods, for both self-supervised and supervised perceptual loss networks. The accuracy is computed as the the %-age of points within a specified pixel distance $d$. In this case, the top two supervised methods are better than our unsupervised approach, but we outperform [33, 53] using 1k training samples (vs. 10k); furthermore, methods such as [35] are specialised for videos and

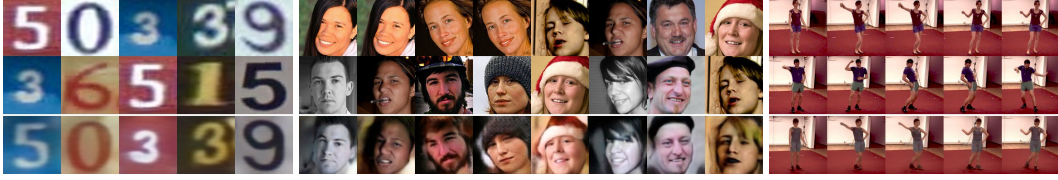

Figure 7: **Disentangling Style and Geometry.** Image generation conditioned on *spatial* keypoints induces disentanglement of representations for style and geometry in the generator. Source image (**x**) imparts style (*e.g.* colour, texture), while the target image (**x'**) influences the geometry (*e.g.* shape, pose). Here, during inference, **x** **[middle]** is sampled to have a different *style* than **x'** **[top]**, although during training, image pairs with *consistent* style were sampled. The generated images **[bottom]** borrow their style from **x**, and geometry from **x'**. **(a) SVHN Digits:** the foreground and background colours are swapped. **(b) AFLW Faces:** pose of the style image **x** is made consistent with **x'**. **(c) Human3.6M:** the background, hat, and shoes are retained from **x**, while the pose is borrowed from **x'**. All images are sampled from respective test sets, never seen during training.

leverage temporal smoothness. Training using the supervised perceptual loss is understandably better than using the self-supervised one. Performance is particularly good on parts such as the elbow.

### 4.4   Learning 3D object landmarks: pose, shape, and illumination invariance

We train our unsupervised keypoint detectors on the SmallNORB [26] dataset, comprising 5 object categories with 10 object instances each, imaged from regularly spaced viewpoints and under different illumination conditions. We train category-specific detectors for $K = 20$ keypoints using image-pairs from neighbouring viewpoints and show results in fig. 6 for *car* and *airplane* (see supplementary material for visualisation of other object categories). Keypoints most invariant to various factors are visualised. These landmarks are especially robust to changes in illumination and elevation angle. They are also invariant to smaller changes in azimuth ($\pm 80°$), but fail to generalise beyond that. Most interesting, they localise structurally similar regions, even when there is a large change in object shape (*e.g.* fig. 6-(d)); such landmarks could thus be leveraged for viewpoint-invariant semantic matching.

### 4.5   Disentangling appearance and geometry

In fig. 7 we show that our method can be interpreted as disentangling appearance from geometry. Generator/ keypoint networks are trained on SVHN digits [29], AFLW faces, and Human3.6M people. The generator network is capable of retaining the geometry of an image, and substituting the style with any other image in the dataset, including unrelated image pairs never seen during training. For example, in the third column we re-render the number 3 by mixing its geometry with the appearance of the number 5. This generalises significantly from the training examples, which only consist of pairs of digits sampled from the *same* house number instance, sharing a common style.

## 5   Conclusions

In this paper we have shown that a simple network trained for conditional image generation can be utilised to induce, without manual supervision, a object landmark detectors. On faces, our method outperforms previous unsupervised as well as supervised methods for landmark detection. The method can also extend to much more challenging data, such as detecting landmarks of people, and diverse data, such as 3D objects and digits.

**Acknowledgements.**   We are grateful for the support provided by EPSRC AIMS CDT, ERC 638009-IDIU, and the Clarendon Fund scholarship. We would like to thank James Thewlis for suggestions and support with code and data, and David Novotný and Triantafyllos Afouras for helpful advice.

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
