[Supplementary Material]

# Unsupervised Learning of Object Landmarks through Conditional Image Generation
## Supplementary Material

**Tomas Jakab**[1][*]     **Ankush Gupta**[1][*]     **Hakan Bilen**[2]     **Andrea Vedaldi**[1]

[1] Visual Geometry Group
University of Oxford
{tomj,ankush,vedaldi}@robots.ox.ac.uk

[2] School of Informatics
University of Edinburgh
hbilen@ed.ac.uk

## Appendix

We first present more detailed results on MAFL dataset comparing performance of different versions of our method. Then we show extended versions of figures presented in the paper. The sections are organized by the datasets used.

## 1   MAFL

| | Training set → | CelebA | | | VoxCeleb |
|---|---|---|---|---|---|
| $K$ landmarks | Regression set | Thewlis [3] | sup. | selfsup. | sup. |
| 10 | | 7.95 | 3.32 | 3.19 | — |
| 30 | MAFL | 7.15 | 2.63 | 2.58 | 4.17 |
| 50 | | 6.67 | 2.59 | 2.54 | 3.59 |
| 10 | | 6.32 | 3.32 | 3.19 | — |
| 30 | CelebA | 5.76 | 2.63 | 2.57 | 4.14 |
| 50 | | 5.33 | 2.59 | 2.53 | 3.55 |

Table 1: **Results on MAFL face-landmarks test-set.** Varying number ($K$) of unsupervised landmarks are learnt on two training-sets — random-TPS warps on CelebA [1], and face-videos from the VoxCeleb [2]. These landmarks are regressed onto 5 manually-annotated landmarks in the MAFL [4] test set, using either CelebA or MAFL training sets. Mean squared-error (MSE) normalised by the inter-ocular distance is reported.

## 2   Boundary Discounting

When TPS warping is used during training some pixels in the resulting image may lie outside the original image. Since reconstructing these empty pixels is irrelevant we ignore them in the reconstruction loss. We additonaly ignore 10 pixels on the edges of the original image and use a smooth step over the next 20 pixels. This is to further discourage reconstruction of the empty pixels as they can influence the perceptual loss when a convolutional neural network with a large receptive field is used.

---

[*]equal contribution.

# 3    MAFL and AFLW Faces

Figure 1: Supervised linear regression of 5 keypoints (bottom rows) from 10 unsupervised (top rows) on MAFL (above) and AFLW (below) test sets. Centre of the white-dots correspond to the ground-truth location, while the dark ones are the predictions. The models were trained on random-TPS warped image-pairs; self-supervised peceptual-loss network was used.

# 4 VoxCeleb

Figure 2: Training with video frames from VoxCeleb. **[rows top-bottom]:** (1) source image $\mathbf{x}$, (2) target image $\mathbf{x}'$, (3) generated target image $\Psi(\mathbf{x}, \Phi(\mathbf{x}'))$, (4) unsupervised landmarks $\Phi(\mathbf{x}')$ superimposed on the target image. The landmarks consistently track facial features.

# 5 BBCPose

Figure 3: **Learning Human Pose.** 50 unsupervised keypoints are learnt. Annotations (empty circles) for 7 keypoints are provided, corresponding to — head, wrists, elbows and shoulders. Solid circles represent the predicted positions; Top rows show raw discovered keypoints which correspond maximally to each annotation; bottom rows show linearly regressed points from the discovered keypoints. **[above]:** randomly sampled frames for different actors **[below]:** frames from a video track.

# 6 Human3.6M

Figure 4: **Unsupervised Landmarks on Human3.6M.** Video of two actors (S1, S11) "posing", from the Human3.6M test set. **(rows)** (1) source, (2) target, (3) generated, (4) landmarks, (5) landmarks on frames from a different view, (6–7) landmarks on two views of the second actor. The landmarks consistently track the legs, arms, torso and head across frames, views and actors. However, the model confounds the frontal and dorsal sides.

# 7 smallNORB 3D Objects: pose, shape, and illumination invariance

Object-category specific keypoint detectors are trained on the 5 categories in the smallNORB dataset — *human, car, animal, airplane,* and *truck*. Training is performed on pairs of images, which differ only in their viewpoints, but have the same object instance (or shape), and illumination.

Keypoints invaraint to viewpoint, illumniation, and object shape are visualised for object instances in the test set. The training set consists of only 5 object instances per category, yet the detectors generalise to novel object instances in the test set, and correspond to structurally similar regions across instances.

elevation →
azimuth →

shape →
illumination →

elevation →
azimuth →

shape →
illumination →

elevation →
azimuth →

shape →
illumination →

# 8 Disentangling appearance and geometry

The generator substitutes the appearance of the target image ($\mathbf{x}'$) with that of the source image ($\mathbf{x}$). Instead of sampling image pairs ($\mathbf{x}, \mathbf{x}'$) with *consistent* style, as done during training, we sample pairs with *different* styles at inference, resulting in compelling transfer across different object categories — SVHN digits, Human3.6M humans, and AFLW faces.

Figure 5: **SVHN digits.** Target, source, and generated image triplets $\langle \mathbf{x}', \mathbf{x}, \Psi(\mathbf{x}, \Phi(\mathbf{x}')) \rangle$ from the SVHN test set. The digit shape is swapped out, while colours, shadows, and blur are retained.

Figure 6: **Human3.6M humans.** Transfer across actors and viewpoints. **[top]:** different actors in various poses, imaged from the same viewpoint; the pose is swapped out, while appearance characteristics like shoes, clothing colour, and hat are retained. **[bottom]:** successful transfer even when the target is imaged from a different viewpoint (same poses as above).

Figure 7: **AFLW Faces.** The source image $\mathbf{x}$ is rendered with the pose from the target image $\mathbf{x}'$; the identity is retained.