[Reviews · NeurIPS 2018]

Reviewer 1



Summary: This paper proposes a method for conditional image generation by jointly learning "structure" points such as face and body landmarks. The authors propose to use a convolutional neural network with a modified loss to capture the image transformation and landmarks. They evaluate their approach on a set of datasets including CelebA, VoxCeleb, and Human 3.6M. Positive: -The problem addressed is an important problem and the authors attempt to solve it using a well engineered approach. -The results seem to be competitive to some state of the art approaches. Negatives: -The pre-processing using heat maps, normalizing them into probabilities, then using a gaussian kernel to produce the features is a bit heuristic. -The claim in P4 L132-135 saying this is usually done using GANs, it would have been good to show some results and comparison against GAN based methods. Comparison against [37] for image generation using landmark/control/structure points would have helped. -The use of two networks, landmark network and image generation network, rather than training a joint network seems inefficient. Landmarks and image generation are two complimentary that could be modeled in a joint network.

Reviewer 2



The paper presents a way to generate a new image from another similar one conditioned on the unsupervised detected landmark points. Although being an interesting way to do style transferring, the techniques used in the work is weakly motivated and did not have major architecture contribution at the level of NIPS. The highlighted distinction of the paper is the learning landmarks together with the generator, but the motivation for doing this is not clear in the text. This limitation involves the undefined meaning of these landmarks and how they represent the “structure” of the image. It would be clearer if the author used correspondence pairs of landmarks in the source and target images. The presentation will need a lot of improvement for further clarity and grammatical correctness. Edit: At a second look at the paper and the rebuttal, I agreed that I have misunderstood the goal of the paper toward image generation. Toward the actual goal of unsupervised learning of landmark points I think the idea is rather neat and creative. I will raise my vote to weak accept hoping that the authors will make the main objective more obvious in their edits.

Reviewer 3



This paper develops an unsupervised approach to learning image landmarks. The key of the proposed model is a representation bottleneck that only allows landmark information of an image to flow through and be fed to the conditional image generator to recover the image. Experiments show the generality of the method on different types of images. The idea looks clean and quite reasonable. The representation bottleneck ensures only the landmark information is distilled from the image, and other irrelevant information is filtered out. Presentation is clear. Fig.1 is informative and helpful. The quality of the example images (e.g., Fig.3 and 4) should be improved. In particular, the landmarks are hard to see.